# Augmenting light coverage for photosynthesis through YFP-enhanced charge separation at the *Rhodobacter sphaeroides* reaction centre

Katie J. Grayson[1], Kaitlyn M. Faries[2], Xia Huang[1], Pu Qian[1], Preston Dilbeck[2], Elizabeth C. Martin[1], Andrew Hitchcock[1], Cvetelin Vasilev[1], Jonathan M. Yuen[2], Dariusz M. Niedzwiedzki[3], Graham J. Leggett[4], Dewey Holten[2], Christine Kirmaier[2] & C. Neil Hunter[1]

Photosynthesis uses a limited range of the solar spectrum, so enhancing spectral coverage could improve the efficiency of light capture. Here, we show that a hybrid reaction centre (RC)/yellow fluorescent protein (YFP) complex accelerates photosynthetic growth in the bacterium *Rhodobacter sphaeroides*. The structure of the RC/YFP-light-harvesting 1 (LH1) complex shows the position of YFP attachment to the RC-H subunit, on the cytoplasmic side of the RC complex. Fluorescence lifetime microscopy of whole cells and ultrafast transient absorption spectroscopy of purified RC/YFP complexes show that the YFP–RC intermolecular distance and spectral overlap between the emission of YFP and the visible-region ($Q_X$) absorption bands of the RC allow energy transfer via a Förster mechanism, with an efficiency of $40 \pm 10\%$. This proof-of-principle study demonstrates the feasibility of increasing spectral coverage for harvesting light using non-native genetically-encoded light-absorbers, thereby augmenting energy transfer and trapping in photosynthesis.

[1] Department of Molecular Biology and Biotechnology, University of Sheffield, Sheffield S10 2TN, UK. [2] Department of Chemistry, Washington University, St. Louis, Missouri 63130-4889, USA. [3] Photosynthetic Antenna Research Center, Washington University, St. Louis, Missouri 63130-4889, USA. [4] Department of Chemistry, University of Sheffield, Brook Hill, Sheffield S3 7HF, UK. Correspondence and requests for materials should be addressed to C.N.H. (email: c.n.hunter@sheffield.ac.uk).

                                                                                                               1

Photosynthesizing organisms harvest only a small proportion of the visible and near-infrared radiation reaching the Earth's surface, and bacterial, algal and plant species have evolved to occupy particular spectral niches depending on the absorption of the particular chlorophyll, carotenoid, phycobilin and other pigments they synthesize[1,2]. Several studies have explored the possibilities of expanding the spectral coverage of light harvesting and reaction centre (RC) complexes using chlorophylls[3–5] or attached dye molecules[6–8] and the biohybrid approach, which combines parts of native antennas with synthetic chromophores, provides an emerging platform technology for the creation of artificial antennas. This flexible, versatile and tailorable strategy extends solar coverage beyond natural systems. The polypeptide scaffold often chosen for such work is based on the light-harvesting LH1 complex from the photosynthetic bacterium Rhodobacter (Rba.) sphaeroides. The LH1 antenna consists of a circular assembly of repeating $\alpha_1\beta_1 BChl_2 Crt_2$ units, where each $\alpha$ or $\beta$ polypeptide is a single transmembrane helix that binds a single bacteriochlorophyll $a$ (BChl $a$) and a carotenoid (Crt) molecule. This assembly surrounds, and donates excitation energy to, the RC where charge separation occurs. The discovery that an LH1 antenna can be reconstituted from purified pigments and $\alpha/\beta$ polypeptides, plus the availability of single-site alterations and truncated versions of these polypeptides, allowed the in vitro self-assembly of LH1 antenna variants[9–11]. Biohybrid antennas with enhanced spectral coverage can be assembled by creating attachment sites for synthetic chromophores that efficiently transfer energy to the BChl $a$ site[5,12–15]. The efficiencies of energy transfer to the BChl $a$ acceptor are comparable with native antennas, and current designs can accommodate up to 60 chromophores that contribute to solar light harvesting via an energy-transfer cascade. The use of unattached accessory chromophores within detergent micelles provides another route to enhance light-harvesting capability[14,15].

Biohybrid architectures have great potential for producing artificial light-harvesting architectures in vitro, but they utilize synthetic chemistry so they cannot replicate in vivo. To create tailor-made light harvesting antennas in vivo, we must make use of the toolbox of proteins and pigments available in nature, or create synthetic elements that can be produced by the host organism. In the purple photosynthetic bacterium Rba. sphaeroides, light is harvested by the LH2 and LH1 antenna complexes and the resulting excitation energy is used to power the reduction of quinone to quinol before the formation of a proton gradient that powers ATP synthesis[16]. Rba. sphaeroides transfers the energy from light absorbed by B875 BChls in LH1 and B850 BChls in LH2 to the RC with near 100% quantum efficiency[17–19]. A charge separation takes place within the RC, which converts excitation energy to biochemical energy[20]. Quinols, the eventual product of RC photochemistry, exit from the RC and traverse the LH1 ring surrounding the RC through a portal provided by the PufX polypeptide[16].

As a pilot study to explore the possibility of creating artificial light-harvesting antennas in vivo, this work investigates the effects of incorporating the yellow fluorescent protein, YFP, as a chromophore into the Rba. sphaeroides photosystem. YFP is a well-studied and genetically programmable protein; the YFP variant used in this study, SYFP2, exhibits photostability, a high-fluorescence quantum yield ($\sim$70%) and a large extinction coefficient ($\sim$100,000 $M^{-1}$ $cm^{-1}$ at 515 nm)[21]. However, YFP cannot contribute usefully to the native absorption of Rba. sphaeroides, because the carotenoid pigments of the light-harvesting complexes absorb in the 400–550 nm range. Accordingly, we constructed a carotenoid-less mutant of Rba. sphaeroides to create a baseline strain for assessing any contribution of YFP to photosynthesis. The 525 nm emission

maximum of SYFP2 has some overlap with the visible-region ($Q_X$) absorption band of BChl $a$ in the Rba. sphaeroides LH1 and RC complexes, and the results of this study suggest that some of the energy absorbed by YFP migrates to the RC. This work forms the basis of future studies involving the creation of novel, tailored light-harvesting antenna complexes for increased spectral coverage of in vivo photosynthesis.

## Results

**Spectroscopy and photosynthetic growth of RC/YFP strains.** A baseline strain was constructed for the expression of YFP by deleting crtB (RSP_0270, encoding phytoene synthase) from Rba. sphaeroides[22]. This manipulation abolishes the biosynthesis of carotenoids, the 450–550 nm absorption of which would otherwise eclipse YFP absorption, as shown in the absorption spectra for WT and WT RC/YFP membranes (Supplementary Fig. 1). The $\Delta crtB$ mutation also eradicates LH2 assembly[23,24] but does not affect the synthesis of the PufX polypeptide, which forms a portal for quinone traffic[16,25]. Although PufX is present, the carotenoid-less RC–LH1–PufX complex with YFP attached will be referred to as $\Delta crtB$ RC/YFP–LH1. The gene encoding the YFP variant, SYFP2 (ref. 21), was fused to the 3′ end of puhA, which encodes the reaction centre H subunit (RC-H). The placement of YFP on the periplasmic N terminus of RC-H was ruled out as, although it would bring the YFP closer to the RC special pair and the LH1 BChls, it would have likely blocked the cytochrome $c_2$-docking site. The result of constructing puhA-SYFP2 is a fusion protein with YFP on the C-terminus of RC-H, located on the cytoplasmic side of the complex.

Immunoblotting with antibodies specific for the RC-H subunit or YFP showed the presence of a signal at 54.9 kDa corresponding to the expected size of a RC-H/YFP fusion (Fig. 1a). Room temperature absorption spectra of intracytoplasmic membranes (ICMs) prepared from photosynthetically grown $\Delta crtB$ RC/YFP–LH1 show the YFP peak at 517 nm, which is shifted to 519 nm at 77 K (Fig. 1b,c). Fluorescence excitation spectra of ICM prepared from $\Delta crtB$ RC/YFP–LH1 recorded at 77 K exhibit a feature at 519 nm corresponding to energy-transfer from YFP (Fig. 1d).

**Photosynthetic growth from absorption of light by YFP.** The 400–550 nm carotenoid absorption of light-harvesting LH1 and LH2 complexes overwhelms the absorption of RC-attached YFP (Supplementary Fig. 1), so WT and WT RC/YFP–LH1 strains grow at the same rate under illumination by white light (Supplementary Fig. 2). Nevertheless, the 525 nm emission maximum of SYFP2 has some overlap with the visible-region ($Q_X$) absorption band of BChl $a$ in the Rba. sphaeroides LH1 and RC complexes, so growth tests were conducted to find out if energy absorbed by YFP could migrate to RC–LH1 complexes in vivo. The carotenoid-less $\Delta crtB$ mutant of Rba. sphaeroides provides the ideal baseline strain for growth rate experiments; the $\Delta crtB$ and $\Delta crtB$ RC/YFP–LH1 strains were grown in white light (100 µmol photons $s^{-1}$ $m^{-2}$), under which conditions both strains are able to grow autotrophically. The LH2-minus positive control (carotenoids + RC–LH1–PufX), chosen because the other two strains also lack LH2 due to the $\Delta crtB$ mutation, shows strong photosynthetic growth (Fig. 2a, green). Allowing for the $\sim$30 h lag delaying the onset of photosynthetic growth, the subsequent difference between the green and blue growth curves shows that under white light illumination the $\Delta crtB$ culture is light-limited by $\sim$50 %. This effect arises because the absence of carotenoids impairs the light-absorbing capacity of the LH1 antenna, thereby limiting turnover at the RCs. After $\sim$30 h the growth of $\Delta crtB$ RC/YFP–LH1 cultures began to outpace the

control $\Delta crtB$ strain, providing evidence for *in vivo* energy transfer from YFP to the RC–LH1–PufX complex (Fig. 2a). In an additional growth test, an LED light source (520 nm emission maximum, 35 nm full-width-at-half-maximum (FWHM)) was used to direct excitation energy specifically to YFP. The difference between the positive control (carotenoids + RC–LH1–PufX; green line) and $\Delta crtB$ strains (no carotenoids, red/blue lines) in Fig. 2c shows that both $\Delta crtB$ and $\Delta crtB$ RC/YFP–LH1 are growing at

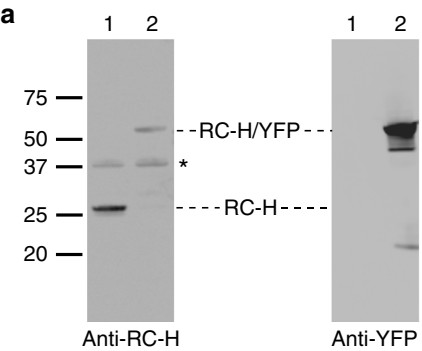

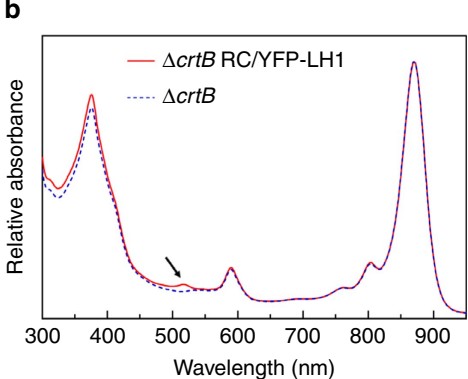

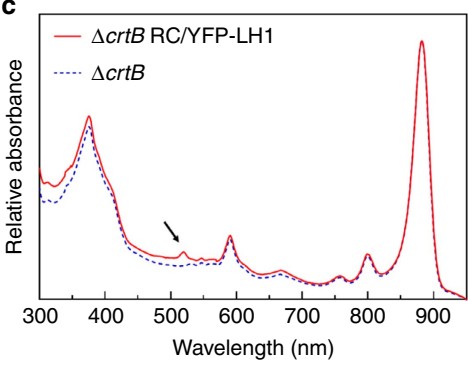

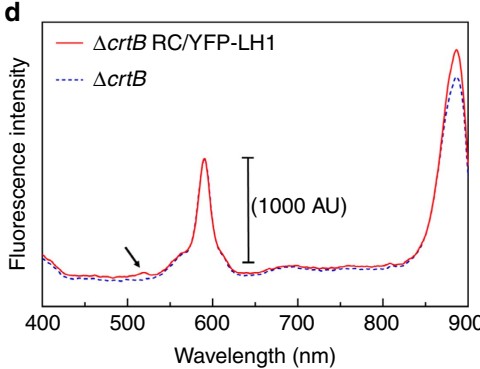

sub-saturating light intensities. Illumination of carotenoid-less cultures by green LEDs barely allows native BChl *a* pigments to contribute towards light harvesting. Comparison of the green and blue growth curves in Fig. 2c suggests that RC turnover in the $\Delta crtB$ strain runs at only ~10% of its potential capacity so the addition of a 520 nm-absorbing chromophore to the $\Delta crtB$ strain might be expected to be beneficial, as long as the absorbed energy can transfer to photosystem components. The emission of the green LEDs (depicted by the white region in Fig. 2b) is well matched to the absorption of YFP, and the $\Delta crtB$ RC/YFP–LH1 strain (Fig. 2c, red) grows faster than $\Delta crtB$ under these experimental conditions. In contrast, the $\Delta crtB$ negative control absorbed enough energy to sustain only very weak photosynthetic growth (Fig. 2c, blue).

The inset in Fig. 2c shows that the growth conditions supplied sufficient oxygen for maturation of the YFP chromophore in the $\Delta crtB$ RC/YFP–LH1 strain. Thus, the data presented in Fig. 2 provide further evidence for light absorption by YFP and transfer of excitation energy to the RC to drive photosynthesis and cell division. However, the absence of carotenoids in the $\Delta crtB$ strain used for this work could have created a need for photoprotective pigments, and it has been suggested that fluorescent proteins can play a photoprotective role[26,27]. To confirm that the superior growth rate of the $\Delta crtB$ RC/YFP–LH1 strain (versus the $\Delta crtB$ negative control) was due to YFP energy transfer and not photoprotection (which could favour an increased growth rate), the expression plasmid pBBRBB–YFP, based on pBBRBB-$Ppuf_{843–1200}$[28], was used to make cytoplasmic YFP in *Rba. sphaeroides* $\Delta crtB$, which is free of any attachment to membrane proteins. Photosynthetic growth curves for high light cultures show that the presence of free YFP in the cytoplasm confers no benefit on $\Delta crtB$ pBBRBB–YFP (Supplementary Fig. 3), which grows at the same slow rate as $\Delta crtB$. The effectiveness of this plasmid expression system for YFP synthesis in *Rba. sphaeroides* was verified by the subsequent purification of YFP from this strain, which was used as a control for the static and time-resolved fluorescence measurements.

**Structural model of the RC/YFP–LH1 complex.** To visualize the attachment of YFP to the RC-H subunit, the RC/YFP–LH1 complex was purified from ICM prepared from photosynthetically grown cells, as described in the Methods. Following sucrose density gradient centrifugation, ion exchange chromatography and gel filtration the final complex was reasonably pure, as judged by the A875:A280 nm absorbance ratio of 1.92 and the presence of only five main bands (the PufX polypeptide stains poorly[29]) in a Coomassie-stained SDS–PAGE gel (Supplementary Fig. 4).

Purified RC/YFP–LH1 complexes were adsorbed onto the pre-coated carbon film on a copper grid and negatively stained; Fig. 3a shows typical raw data with some single particles showing

**Figure 1 | Immunoblotting and spectroscopic analysis of $\Delta crtB$ RC/YFP–LH1 ICMs.** (**a**) Immunoblotting with antibodies for the RC-H subunit and, separately, to YFP showing the synthesis of an RC-H/YFP polypeptide in *Rba. sphaeroides*. Lanes 1 and 2 indicate the $\Delta crtB$ and $\Delta crtB$ RC/YFP–LH1 strains, respectively. The asterisk indicates a non-specific signal from the RC-H antibodies. The numbers to the left of the anti-RC-H blot show the positions of protein standards, in kDa. (**b**) Room temperature and (**c**) 77 K absorption spectra of membranes purified from $\Delta crtB$ (blue, dashed) and $\Delta crtB$ RC/YFP–LH1 (red) normalized to 870 nm (**b**) or 880 nm (**c**). $\Delta crtB$ RC/YFP–LH1 has a peak at 517 nm corresponding to YFP (indicated by arrow); this peak is shifted to 519 nm at 77 K. (**d**) Fluorescence excitation spectra of membranes with emission monitored at 910 nm show that YFP (peak indicated by arrow) contributes to emission at 910 nm.

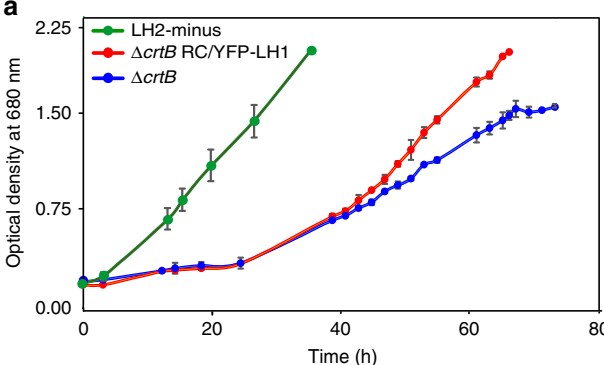

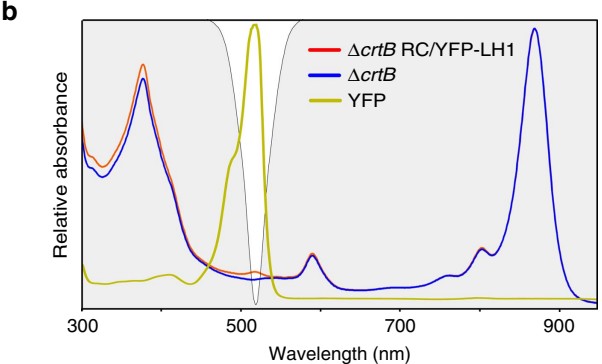

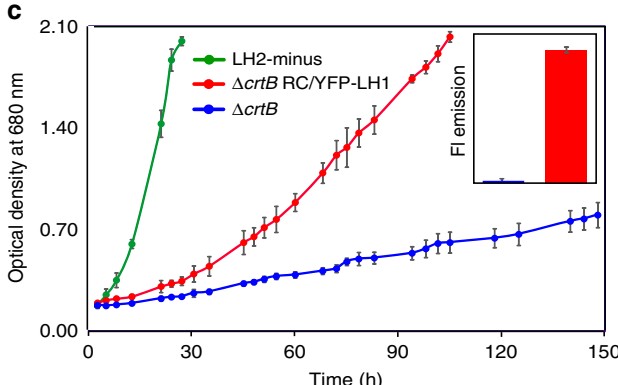

**Figure 2 | Photosynthetic growth curve analysis of ∆crtB RC/YFP–LH1.**
(**a**) Light was provided using Megaman CFL bulbs at an intensity of
100 µmol photons s$^{-1}$ m$^{-2}$. Photosynthetic growth curves are shown for
∆crtB RC/YFP-LH1 (red circles) and ∆crtB (blue circles); a carotenoid-
containing LH2-minus strain (green circles) is included as a positive control.
(**b**) Superposition of the absorption of ∆crtB, ∆crtB RC/YFP-LH1 and purified
YFP onto the emission output of the LEDs used for the photosynthetic growth
rate experiment in **c**. ∆crtB is shown in blue, ∆crtB RC/YFP-LH1 in red and
purified YFP in yellow; the spectra are normalized to their maximum values.
The emission of the green LEDs (520 nm emission maximum, 35 nm FWHM)
is indicated by the white region on the grey background. (**c**) Photosynthetic
growth curves of ∆crtB RC/YFP-LH1 (red circles) and ∆crtB (blue circles)
under green LEDs. A carotenoid-containing LH2-minus strain (green circles)
is included as an additional control. Error bars indicate the s.d. from the mean;
$n = 3$. Each growth curve is representative of at least three independent
experiments. The inset shows a quantification of YFP fluorescence,
normalized for equal numbers of cells, showing that there was sufficient
oxygen for maturation of the YFP chromophore.

an additional density, indicated by white arrows in the inset
to Fig. 3a. Then 11,777 particles were selected for further analysis;
Fig. 3b shows examples of averaged reference-free 2D classes and

Fig. 3c,d shows the reconstructed model of the RC/YFP–LH1
complex viewed from the cytoplasmic side of the membrane and
in the plane of the membrane, respectively. This model is based
on half of the structure of the dimeric RC–LH1–PufX core
complex[29,30]; the extra density apparent in some of the single
particles and image classes and seen in the final reconstruction
arises from YFP, which clearly protrudes from the side of the
complex.

**YFP fluorescence lifetimes in *Rba. sphaeroides* cells.** A home-
built microscope[31] was used to image fluorescence from YFP,
both in the free state in ∆crtB pBBRBB–YFP cells, and attached to
RC-H in the ∆crtB RC/YFP-LH1 strain; the ∆crtB strain was
included as a negative control. The images in Fig. 4a,c show the
YFP emission from YFP in the RC-attached and free states,
respectively. Figure 4d shows fluorescence emission spectra, each
recorded on a single cell; the shapes of the spectra match those for
purified YFP and RC/YFP–LH1 samples in Fig. 5b. No
conclusions can be drawn from the amplitudes, which reflect
the relative abundance of YFP in the free and attached states in
these particular cells, as well as the fluorescence yield arising from
coupling to acceptors in the RC/YFP–LH1 complex. As expected,
no signal was observed in cells of the control ∆crtB mutant,
(Fig. 4b,d, black trace). Fluorescence lifetimes of YFP in the whole
cell samples were recorded to detect any quenching that would
indicate Förster resonance energy transfer (FRET) from the YFP
to an acceptor molecule (Fig. 4e; Table 1). The YFP fluorescence
signal from ∆crtB pBBRBB-YFP, which is an average taken from
eight cells, has a bi-exponential decay (Fig. 4e, black points and
fitted curve) which gives an amplitude-weighted lifetime of 2.3 ns.
When YFP is attached to RCs in ∆crtB RC/YFP–LH1 cells the
lifetimes of both the short- and long-lived components are
significantly shorter (Fig. 4e, red points), with an average lifetime
of 1.2 ns. These amplitude-weighted lifetimes (Table 1) and
Equation 1 were used to calculate a FRET efficiency ($\Phi_{EET}$) for
energy transfer between YFP and the RC–LH1 complex of ∼0.48.
Figure 3f shows the distances between the YFP chromophore and
the RC special pair BChls, and between YFP and the nearest LH1
B875 BChl, based on the structural model of the RC/YFP–LH1
complex in Fig. 3c,d. These distances, of ∼85 Å and ∼53 Å,
respectively, are both within the range for FRET, although they
suggest that the most likely route for energy migration to the RC
is via the LH1 BChls.

$$\Phi_{EET} = 1 - \frac{\tau_{YFP-RC}}{\tau_{YFP}} \qquad (1)$$

To examine this point in more detail we used Förster theory to
estimate the distance between the YFP chromophore and
acceptor pigments in the RC–LH1 complex, on the basis of the
0.48 value for $\Phi_{EET}$ determined from the whole cell fluorescence
lifetime measurements. The calculations employed Photo-
chemCad[32] using three values of the refractive index (1.335,
1.45 and 1.55) to accommodate latitude in the polarity of the
cofactor/protein environments (Supplementary Fig. 5). The
calculations return a YFP to acceptor distance of 43–48 Å, in
good agreement with the structural model in Fig. 3f which gives a
53 Å distance between YFP and the nearest LH1 BChl.

**RC/YFP fluorescence quantum yield and excited-state lifetime.**
Having shown that YFP is a donor of excitation energy to RC–
LH1 complexes *in vivo*, we set out to construct the simplest
system for examining the characteristics of a coupled YFP/
energy-trapping complex. Accordingly, RC/YFP complexes with
no surrounding LH1 antenna were purified from carotenoid-less
ICMs. Supplementary Figure 6 shows the outcome of the

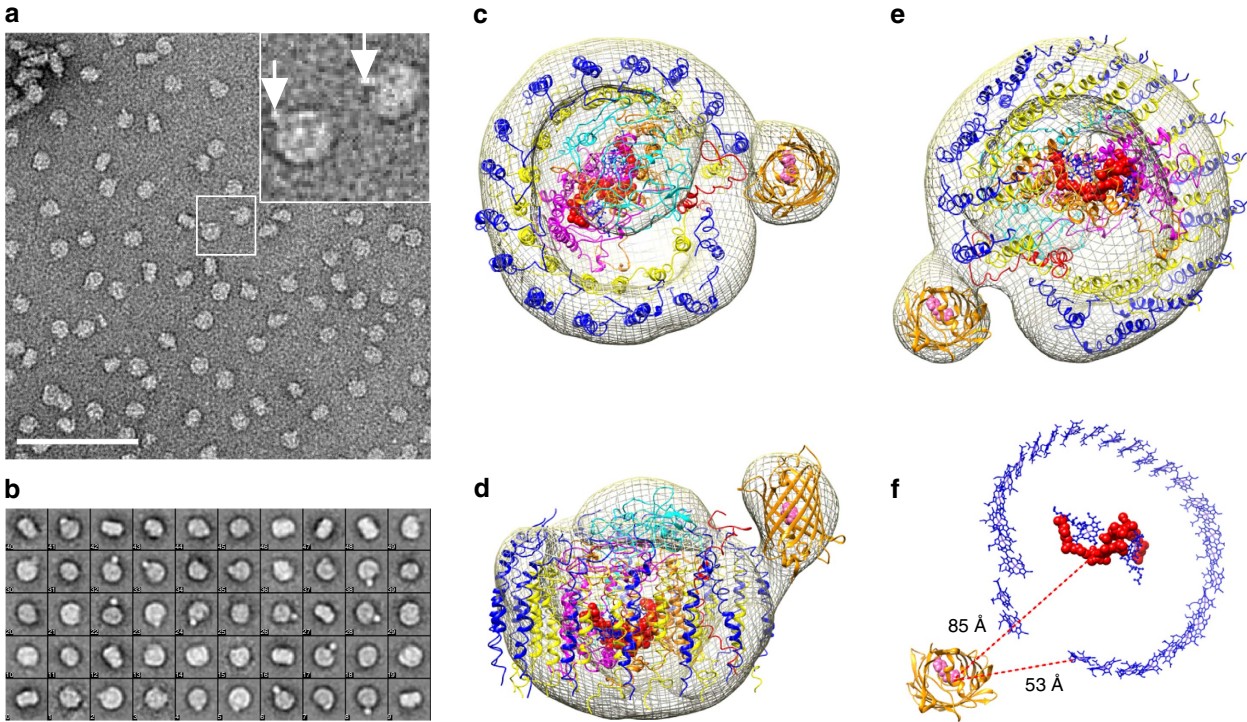

**Figure 3 | Electron microscopy and 3D reconstruction of the RC/YFP-LH1 complex.** (**a**) Electron micrograph of purified, negatively stained RC/YFP–LH1 complexes; the two complexes in the central white box are enlarged in the inset, with white arrows indicating the density arising from YFP. Scale bar, 100 nm. (**b**) Gallery of 40 selected 2D averaged classes; the box size is 28 × 28 nm. Overall, 11,777 particles were used for reconstruction of the 3D model. (**c–f**) The reconstructed model of the RC/YFP–LH1 complex at 29 Å resolution; the diagram shows the electron density (grey mesh), and the following subunits: LH1β (blue), LH1α (yellow), PufX (red), RC-M (magenta), RC-H (cyan) and RC-L (orange). YFP is in gold. The RC special pair of BChls is shown in red. The structural model is viewed from (**c**) the cytoplasmic side of the membrane, (**d**) the plane of the membrane, (**e**) the periplasmic side, (**f**) the periplasmic side of the complex, showing only YFP and the LH1 and RC pigments. The dashed red lines show the distances between the YFP chromophore, the RC special pair BChls (red) and LH1 BChls (blue).

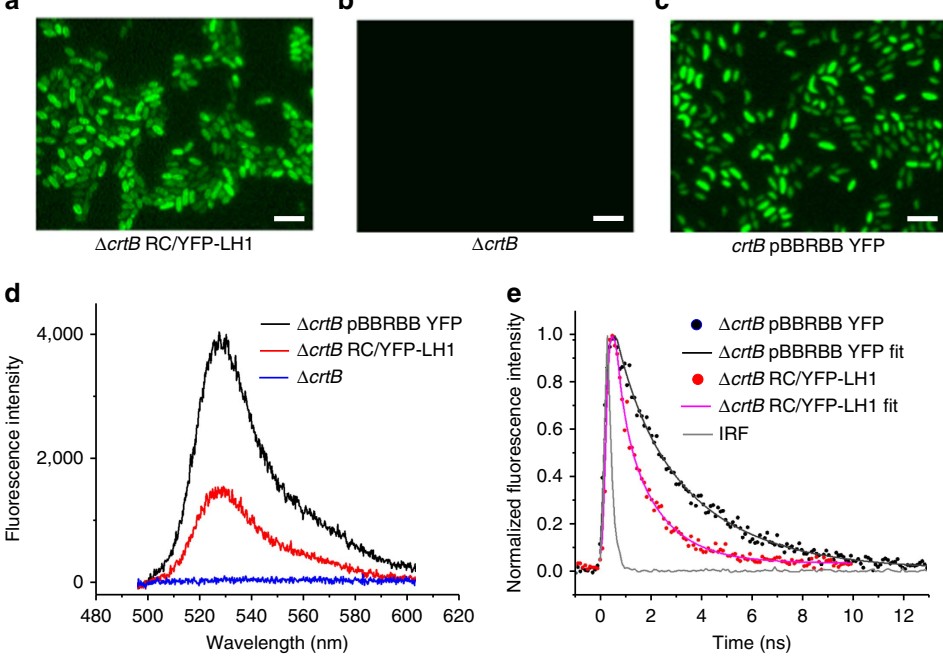

**Figure 4 | Spectral and lifetime imaging of YFP in ΔcrtB RC/YFP-LH1 and ΔcrtB pBBRBB–YFP whole cells.** (**a–c**) Fluorescence images of whole cells of ΔcrtB RC/YFP-LH1, ΔcrtB and ΔcrtB pBBRBB–YFP cells when excited at 495 nm. (**d**) Fluorescence emission spectra, each recorded on a single cell. (**e**) Fluorescence lifetime decay curves recorded at a central wavelength of 550 nm. The best fits, displayed as blue and red lines, were achieved using a double-exponential decay function. The measured instrument response function (IRF) of the system was ∼0.18 ns and taken into account during fitting. Scale bars, 5 μM.

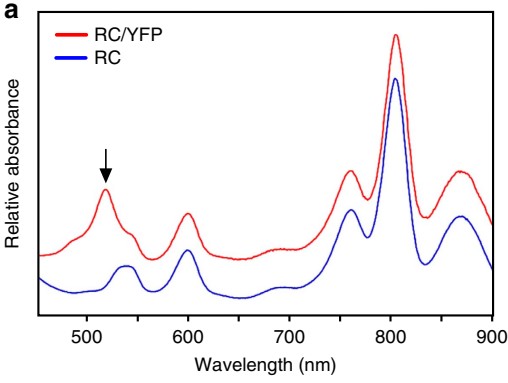

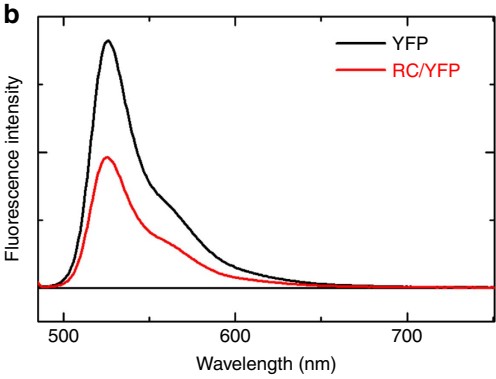

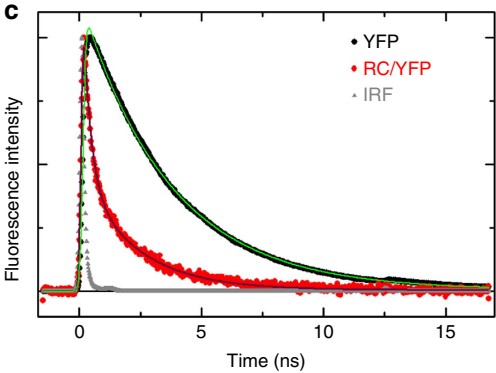

**Figure 5 | Spectroscopic analysis of purified RC/YFP complexes.**
(**a**) Room temperature absorption spectra of control RC complexes purified from the ΔcrtB strain and RC/YFP complexes purified from the ΔcrtB RC/YFP–LH1 strain. The arrow indicates the absorption maximum of YFP at 514 nm. (**b**) Fluorescence spectra of purified YFP (YFP-only; red line) and the RC/YFP complex (black line) normalized to the absorbance at $\lambda_{exc} = 486$ nm. (**c**) Normalized fluorescence decay (at 550 nm using $\lambda_{exc} = 515$ nm) and fits (using two exponentials plus a constant) of YFP-only (black) and RC/YFP (red) obtained via TCSPC.

purification, which yielded complexes free of LH1; Fig. 5a shows the clearly visible YFP absorption at 514 nm (black arrow), which is not present in the RC control. YFP was purified from *Rba. sphaeroides* ΔcrtB pBBRBB–YFP (Supplementary Fig. 7), to act as a control for the fluorescence decay measurements.

The yield of excitation energy transfer ($\Phi_{EET}$) from excited YFP (YFP*) to the RC is determined from changes in the photophysical properties of YFP – fluorescence intensity ($I_f$), fluorescence quantum yield ($\Phi_f$), or singlet excited-state lifetime ($\tau_s$) – in the presence versus absence of the RC. The YFP* → RC energy-transfer process provides an additional decay pathway for YFP* when the RC is present, resulting in a reduction in the measured parameter $\chi$ ($I_f$, $\Phi_f$, $\tau_s$) so that $\Phi_{EET} = 1 - \chi_{RC/YFP}/\chi_{YFP}$. YFP* → RC energy flow was initially measured using static and time-resolved fluorescence spectroscopy. A significant reduction in integrated YFP fluorescence intensity for RC/YFP (Fig. 5b, solid black line) versus YFP-only (red line) gives $\Phi_{EET} = 0.47$. Similarly, as expected based on the simple fluorescence-intensity reduction, the $\Phi_f$ of YFP in RC/YFP is lower than in YFP-only as measured relative to a standard (0.40 versus 0.76) or with an integrating sphere (0.27 versus 0.64), giving $\Phi_{EET}$ values of 0.47 and 0.58, respectively. Finally, time-resolved studies of the YFP fluorescence decay in RC/YFP versus YFP-only by time-correlated single photon counting (TCSPC; Fig. 5c, red and black traces) give $\tau_s$ of 2.0 and 3.6 ns, and a value for $\Phi_{EET} = 0.44$. Analogous YFP fluorescence decays obtained with a stroboscopic method (not shown) give $\tau_s$ of 1.8 and 2.6 ns and thus $\Phi_{EET} = 0.31$.

The $\Phi_{EET}$ values obtained by probing the various photophysical parameters using different techniques are in good agreement and give an average $\Phi_{EET} = 0.4 \pm 0.1$. Similarly the YFP* lifetimes in the RC/YFP complex from the various time-resolved studies (2.2, 2, 1.8 ns) are in good agreement, with an average $\tau_s = 2 \pm 0.2$ ns. These values give a time constant (inverse of the rate constant) for YFP* → RC energy transfer of (2 ns)/0.4 = 5 ns (Fig. 5c).

As in Supplementary Fig. 5, the measured $\Phi_{EET} = 0.4$ was used to estimate the effective distance between the YFP chromophore and the RC cofactors using Förster theory and the photophysical parameters for the energy donor (YFP) and acceptor (RC) given in the 'Methods' section (Supplementary Fig. 8). The calculations return a spectral-overlap integral of $J = 2.96 \times 10^{-13}$ cm⁶, a Förster radius of $R_0 = 53$–58 Å, and a YFP to RC distance of 56–62 Å. The discrepancy between this and the 85 Å distance in Fig. 3d are discussed later.

**Ultrafast transient absorption spectroscopy of RC/YFP.** Ultrafast transient absorption (TA) spectroscopy was used to measure the $\tau_s$ of YFP, and to demonstrate YFP* → RC energy transfer from the 'growing in' of key RC features (absorption bleaching or stimulated emission) associated with formation of the excited primary donor (P*), which initiates the charge-separation sequence in the RC. Figure 6a shows representative TA spectra in

**Table 1 | YFP lifetimes measured for ΔcrtB RC/YFP–LH1 and ΔcrtB pBBRBB–YFP whole cells.**

| Sample | $A_1$ | $\tau_1$ (ns) | $A_2$ | $\tau_2$ (ns) | $\tau_{av}$ (ns) |
|---|---|---|---|---|---|
| ΔcrtB RC/YFP–LH1 | 0.49 ± 0.18 | 1.76 ± 0.28 | 0.51 ± 0.18 | 0.57 ± 0.26 | 1.2 |
| ΔcrtB pBBRBB–YFP | 0.65 ± 0.14 | 3.15 ± 0.39 | 0.35 ± 0.14 | 0.83 ± 0.16 | 2.3 |

The values of amplitudes and lifetimes presented are the mean and s.e., $n = 8$. The fluorescence decay curves shown in Fig. 4e were analysed using OriginPro and TRI2 software packages. $A_1$ and $A_2$ are the amplitude contributions of the long- and short-lived components, $\tau_1$ and $\tau_2$ are the lifetimes of each component, and $\tau_{av}$ is the amplitude-weighted lifetime.

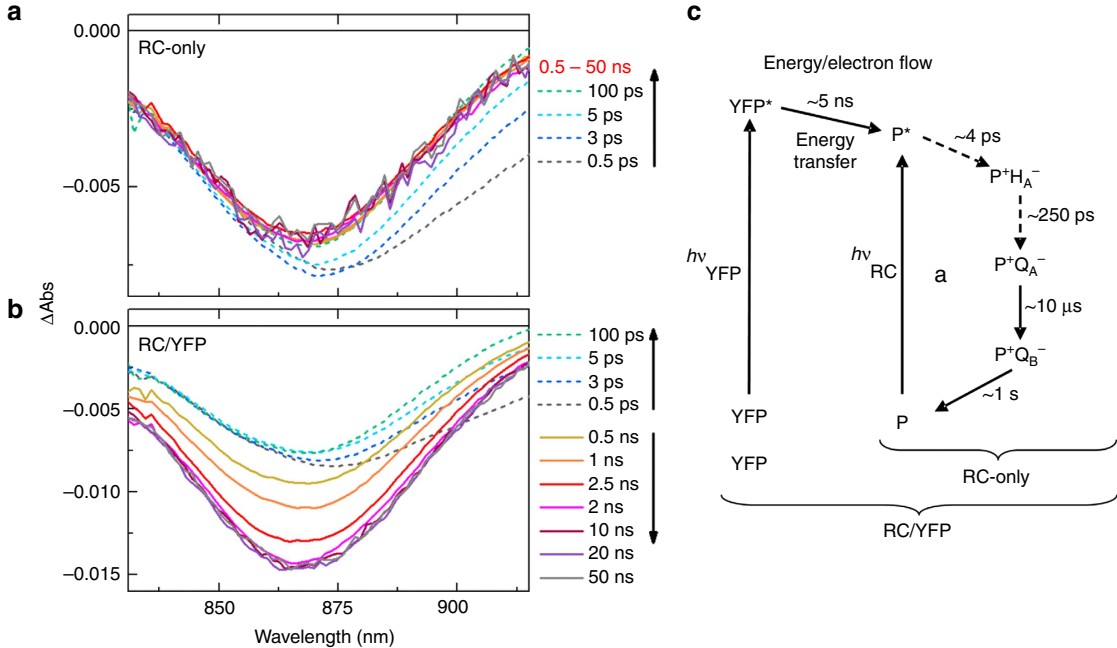

**Figure 6 | Transient absorption spectra of P bleaching at ∼865 nm and P\*-stimulated emission at ∼910 nm for RC-only and RC/YFP samples with 515 nm excitation into the YFP absorption band.** The dashed lines indicate absorbance changes due to P\* (and decay to $P^+H_A^-$ and $P^+Q_A^-$) on direct excitation (at 515 nm) of the RC-only sample (**a**) and in RC/YFP, for which YFP is primarily excited by the flash (**b**). Note the difference in vertical scales for (**a,b**), such that the amplitude of the absorbance changes at early times (dashed) due to direct excitation of the RC are the same in (**a,b**) because the samples have approximately the same concentration. (**c**) Dynamics of energy flow from YFP\* to the RC in RC/YFP followed by charge separation in the RC (in both RC/YFP and RC-only). Dashed and solid arrows correspond to the timescales listed in Figures 6a and 6b.

the near-infrared region for the isolated RC (RC-only) sample, with 515 nm excitation directed at the YFP chromophore. The spectral evolution at early times (dashed lines) arises primarily from decay of P\*-stimulated emission centred at ∼910 nm, with little change in the bleaching of the ground state absorption band of P centred at ∼865 nm. After P\* decay (solid lines) by charge separation, the P bleach persists because P is oxidized ($P^+$) and remains so and in the same amount during the entire charge-separation sequence, namely $P^* \rightarrow P^+H_A^- \rightarrow P^+Q_A^- \rightarrow P^+Q_B^-$. This process proceeds with essentially unity quantum yield; charge recombination $P^+Q_B^- \rightarrow P$ occurs on the time scale of ∼1–10 s (Fig. 6c). Kinetic profiles at 850 and 910 nm were fit to two exponentials plus a constant, giving average time constants of 3.9 and 306 ps, nicely matching typical wild-type RC P\* (∼4 ps) and $P^+H_A^-$ (∼250 ps) lifetimes[33]. As expected, the change in ΔAbs at 850 nm with time is small (nonzero because $H_A^-$ in $P^+H_A^-$ and the other states differ in the size of a small absorption opposing the P bleach) and remains virtually constant from ∼1 ns to past ∼60 ns, the longest time shown (Fig. 7, blue triangles).

TA spectra for the RC/YFP sample at early times (Fig. 6b, dashed lines) give spectral changes with time constants (5.3 and 331 ps) that closely resemble those for RC-only as a result of unavoidable direct RC excitation in a small fraction of the sample. However, at later times (>100 ps), the P bleaching at ∼865 nm triples in magnitude and P\*-stimulated emission at ∼910 nm reappears (Fig. 6b, solid lines). These signals arise unambiguously from YFP\*→RC energy transfer on the nanosecond timescale, which is in marked contrast to the data for RC-only after a few hundred picoseconds (Fig. 6a). For RC/YFP, kinetic profiles at 850 nm (Fig. 7, red circles) and 910 nm were fit to three exponentials plus a constant to account for the RC P\* and $P^+H_A^-$ lifetimes plus a (rise) component reflecting the YFP\* lifetime, which includes YFP\*→RC energy transfer. The $\tau_s$ for

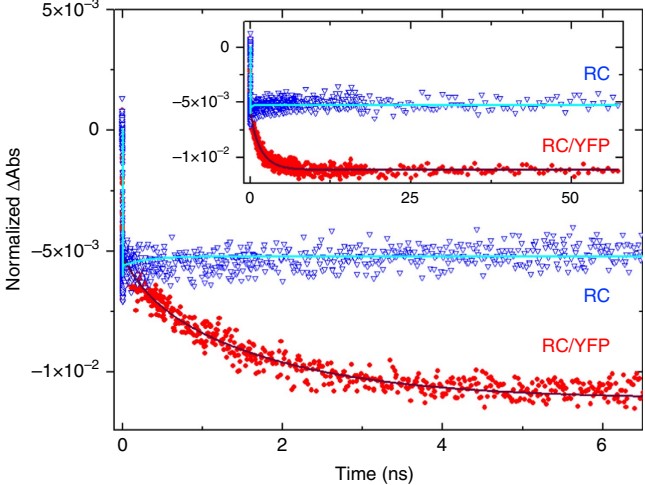

**Figure 7 | Kinetic profiles and fits for evolution of RC P bleaching at 850 nm for RC/YFP and RC-only with 515 nm excitation of YFP.** The signals were normalized at the P absorbance maximum at 867 nm. RC/YFP data (red) was fit to three exponentials plus a constant (solid grey line) and RC-only data (blue, open triangles) to two exponentials plus a constant (cyan).

YFP\* in the RC/YFP complex was determined to be 1.8 ns at 850 nm (Fig. 7, red circles), 2.1 ns at 910 nm from the growth of the RC signals, and 2.4 ns from the decay of YFP-stimulated emission at 560 nm (data not shown), giving an average of 2.2 ns. The YFP stimulated-emission decay at 560 nm in the YFP-only sample gives $\tau_s = 3.4$ ns. The YFP\* decay in the presence/absence of the RC gives an energy-transfer yield of $\Phi_{EET} = 0.35$.

## Discussion

Photosynthetic organisms—plants, algae, cyanobacteria, purple bacteria—are demonstrably successful in occupying their particular spectral niches, but there is scope nevertheless for enhancing the spectral coverage of photosynthesis by augmenting the chlorophyll, carotenoid and phycobilin pigments normally employed for absorbing light. Future synthetic biology applications such as light-powered bioreactors could require the design and construction of organisms that can absorb light over a range of wavelengths not found for naturally-evolved antennas.

Antenna modification *in vitro* can be achieved by attaching highly-absorbing dye molecules to purified complexes, thereby filling gaps in the native absorption spectrum. These biohybrid complexes exhibit efficient energy transfer from attached chromophores to native pigments, and by attaching several chromophores it has been possible to create energy transfer relays[5,12–15]. Attaching up to nine Alexa Fluor 647 molecules to the LH2 complex from *Rhodopseudomonas acidophila* strain 10050 enhanced its light harvesting functions by occupying a vacant spectral region from 600 to 700 nm[8]. Dutta *et al.*[6] have shown that DNA-templated dye molecules can be attached to *Rba. sphaeroides* RCs *in vitro*; the spectral range of these dyes complemented the native absorption of the RCs, and dye-RC energy transfer was observed, with efficiencies ranging from 42 to 83%. In another study, Alexa Fluor dyes were attached directly to the periplasmic face of the RC and dye-RC energy-transfer efficiencies of 52 to 70% were obtained[7].

The present study was performed to determine whether light harvesting and energy transfer could be augmented *in vivo* by genetically modifying a photosystem complex so that it carries an extra chromophore. Photosystems normally contain hundreds of pigments that feed absorbed energy into RC traps, and the attachment of YFP to the RC adds an extra chromophore to the 'photosynthetic unit', in a spectral region already occupied by carotenoids. Thus, it was necessary for this pilot study to make some prior alterations to the bacterial chassis by eliminating the LH2 antenna and carotenoids, clearing space in spectral terms to allow assessment of any contribution from YFP.

In this study we have shown that YFP, when fused to the RC of *Rba. sphaeroides*, is able to transfer electronic excitation energy to the native RC. This process has been observed at three levels; in bacterial cultures, in single cells, and in purified complexes (Figs 1, 2 and 4–7). Attachment of YFP to the RC increases the photosynthetic growth rate of a carotenoid-less mutant of *Rba. sphaeroides* (compared with a control strain with no YFP) when YFP-specific illumination is used. Although the most straightforward explanation of this result is energy transfer from YFP to LH1 and RC pigments (supported by the static and time-resolved microscopy and spectroscopy) we checked for the possibility that YFP mitigates the photosensitivity of the carotenoid-less Δ*crtB* background strain[34,35] since it has been proposed that fluorescent proteins may have a photoprotective function[26,27]. We found that the Δ*crtB* pBBRBB–YFP control strain with cytosolic YFP present, unattached to the RC, did not grow significantly faster than Δ*crtB* and therefore we discount any photoprotective effect of YFP.

We attribute differences in growth rates between carotenoid-containing and carotenoid-less strains (Fig. 2) to the lack of appropriate pigments for absorbing light at 520 nm and show that the attachment of YFP to RCs partly alleviates this problem. Other factors could potentially contribute to the relatively slow growth of carotenoid-less strains, such as destablization of the photosynthetic apparatus or susceptibility to photodamage. We have shown that only the assembly of the LH2 antenna is affected by inactivation of carotenoid biosynthesis[22–25]; thus, we

genetically removed LH2 complexes in the positive control so all strains could be compared on an equal basis in terms of antenna composition. In terms of photodamage, respiration by these photoheterotrophic bacteria during illumination alleviates phototoxic effects in carotenoid-less mutants. Thus, under white light illumination, even the Δ*crtB* mutant achieves good rates of photosynthetic growth even with respect to the carotenoid-containing positive control.

Direct observation of YFP → RC–LH1 energy transfer in whole cells, visualized using fluorescence lifetime microscopy (Fig. 4), is consistent with the increased photosynthetic growth rate of Δ*crtB* RC-H–YFP compared with Δ*crtB* (Fig. 2a–c); the *in vivo* efficiency of energy transfer, $\Phi_{EET}$, was calculated to be 0.48. To discover a structural basis for *in vivo* energy transfer, we used single particle reconstruction to obtain a structural model of the RC/YFP–LH1 complex (Fig. 3). The use of negative stain limited the resolution of the structural model to ∼26 Å, but the presence of YFP was evident from the outset, even in raw images, and the final structural model clearly shows a density that can be confidently assigned to YFP. The staining procedure could affect the perceived orientation of YFP with respect to the rest of the complex; we used over 11,000 particles for calculating the structural model so Fig. 3 shows an averaged orientation of YFP. The use of cryo-EM would produce a better quality structural model, which would also allow a more precise estimate of the distance between the YFP chromophore and the RC special pair. Nevertheless, the low resolution data are sufficient for a calculation of an ∼85 Å distance between the YFP chromophore and the RC special pair and 53 Å to the nearest LH1 B875 BChl (Fig. 3f). Our energy-transfer data provide another way to estimate distances and Förster calculations (Supplementary Fig. 5) show that the acceptor of excitation energy lies 43–48 Å from the YFP donor, consistent with our 53 Å estimate from the structure of the RC/YFP–LH1 complex. Thus, the most likely route for energy transfer from YFP to the RC *in vivo* is via the LH1 BChls.

To make a minimal energy-transfer unit free of LH1, we purified a RC/YFP complex and quantified its energy-transfer dynamics using static and time-resolved absorption and fluorescence spectroscopy. The TA measurements for the RC/YFP complexes (Figs 6 and 7) show an additional ∼1.8 ns component indicating YFP* → RC energy transfer, which is not seen in the RC-only control complex. The energy-transfer yield, $\Phi_{EET}$, was calculated to be 0.35. Measurements of fluorescence yields for RC/YFP and YFP-only by two methods gave 0.47 and 0.58, and fluorescence lifetimes for RC/YFP–RC versus YFP-only using two different methods gave $\Phi_{EET} = 0.44$ and 0.31. The average value for $\Phi_{EET}$ is $0.4 \pm 0.1$. We calculated a 5 ns time constant for YFP* → RC energy transfer and a YFP to RC special pair distance of 56–62 Å (Supplementary Fig. 8), significantly lower than the 85 Å estimate from the RC/YFP–LH1 structure (Fig. 3f). There is a likely structural basis for this apparent discrepancy; the region connecting the YFP and RC-H could accommodate some movement of the YFP in the absence of LH1, allowing YFP to move nearer to the RC 'special pair' BChls (P) and resulting in more efficient energy transfer. This possible change in YFP position, caused by the absence of LH1, is depicted in Fig. 8. The YFP–RC distance of 45 Å measured from this model is shorter than calculated from energy-transfer data, but the model represents the minimum possible YFP–RC distance and in practice the YFP likely does not approach the RC so closely, not least because some detergent molecules likely surround the central part of the RC complex.

In summary, this proof-of-principle study opens up possibilities for the creation of new energy-trapping photosynthetic pathways; we show that translational fusions of

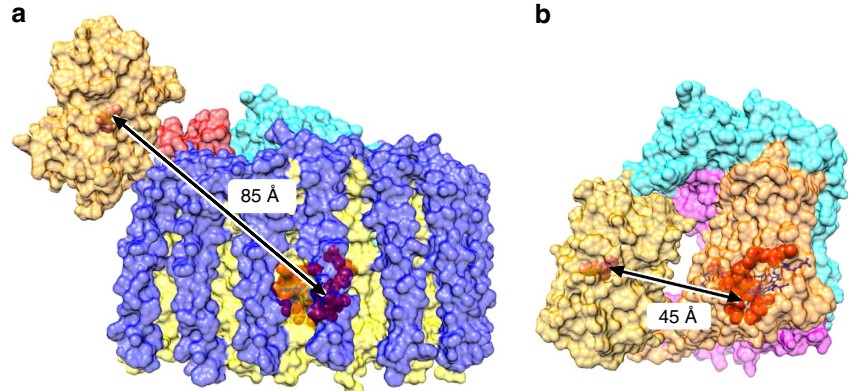

**Figure 8 | Modelling of the effect of LH1 removal on the position of YFP and the energy-transfer distances.** (**a**) The RC/YFP–LH1 complex shown is adapted from the data in Fig. 3. The subunits are: LH1β (blue), LH1α (yellow), PufX (red), RC-M (magenta), RC-H (cyan) and RC-L (orange). YFP is in gold, with its chromophore in orange. The RC special pair of BChls is shown in red. The 85 Å distance between the YFP chromophore and the RC BChl dimer in **a** is shown to shorten when the surrounding LH1 complex is removed during purification of the RC, down to a minimum of 45 Å in **b**.

genetically-encoded light-absorbers and antenna or RC complexes can augment energy transfer and trapping in photosynthesis.

## Methods

**Construction of mutant *Rhodobacter sphaeroides* strains including the ΔcrtB deletion strain.** A construct containing the upstream and downstream flanking regions of the gene was created to delete *crtB*. A 361 bp fragment upstream of *crtB* and the first 7 bp of the *crtB* gene flanked with *Eco*RI and *Xba*I restriction sites (underlined) was amplified using the primers crtBKOUF (5′-CCGGAATTCCA CATCACCATCACCACGGCG-3′) and crtBKOUR (5′-GCGCTCTAGA GATCTAGGTTCTCATGAAGGTATACCG-3′). A second fragment was amplified using the primers crtBKODF (5′-GCGCTCTAGAGGCAATCAT TCCGCGGCAAGC-3′) and crtBKODR (5′-CCCCGCATGCGGCTGTGGCCGA GCCCTA-3′) producing a downstream flanking fragment of 363 bp, which included 9 bp at the end of the *crtB* gene, flanked with *Xba*I and *Sph*I restriction sites (underlined). Following digestion with the indicated enzymes, the fragments were sequentially cloned into the same restriction sites of the suicide vector pK18mobsacB. The resulting plasmid was introduced into *Escherichia coli* S17-1 competent cells and transferred to *Rba. sphaeroides* WT via conjugation, with first and second recombination events selected for as detailed in ref. 31. Transconjugant *Rba. sphaeroides* colonies were selected on M22 agar plates with 25 μg ml⁻¹ kanamycin. Single colonies were grown and scaled up to an 80 ml semi-aerobic culture. The cells were serially diluted 10⁻², 10⁻³ and 10⁻⁴ onto M22 agar containing 1 % (w/v) sucrose and incubated until single colonies appeared after 4–6 days. Single colonies were replica plated onto M22 sucrose plates, with and without 25 μg ml⁻¹ kanamycin. Colonies that grew on the antibiotic-free plate but not on the kanamycin plate were analysed by PCR to identify successful mutants. Transconjugants were confirmed by DNA sequencing; the modified DNA sequence at the location where *crtB* is deleted is shown in Supplementary Fig. 9.

The RC/YFP fusion strain was created using a pk18mobsacB construct designed to fuse the gene for SYFP2 (ref. 21) to the 3′ end of the *puhA* gene encoding the RC-H subunit. To create a C-terminal YFP fusion protein, firstly a PCR fragment was generated using primers puhAYFPUF (5′-CCGGAATTCTCGGCCGGCAAG AACCCGATCGG-3′) and puhAYFPUR (5′-GCTCCTCGCCCTTGCTCACCAT GGCGTATTCGGCCAGCATGGCC-3′). This fragment contained the last 354 bp of the *puhA* ORF (excluding the stop codon), with a 5′ *Eco*RI restriction site (underlined) and 3′ sequence complementary to the start of the *syfp2* gene (italics). A second PCR with primers puhAYFPFor (5′-CGGCGATGCTGGCCGAATACGCC ATGGTGAGCAAGGGCGAGGAGCTGTTCAC-3′) and puhAYFPRev (5′-GCGC TCTAGATCATTACTTGTACAGCTCGTCCATGCCGAGAGTGAT-3′) amplified the *syfp2* ORF with 5′ sequence complementary to the end of the *puhA* gene (italics) and a 3′ *Xba*I restriction site (underlined). The two fragments were used as a template for overlap extension (OLE)-PCR with primers puhAYFPUF and puhAYFPRev, generating a fragment in which the end of the *puhA* gene was joined in frame with the *syfp2* gene with *Eco*RI and *Xba*I sites at the 5′ and 3′ ends respectively. A third fragment in which the 354 bp immediately downstream of the *puhA* stop codon was flanked by *Xba*I and *Hin*dIII restriction sites (underlined) was amplified using the primers puhAYFPDF (5′-GCGCTCTAGATCCCCGCAT GGCGCGGCCC-3′) and puhAYFPDR (5′-CCCCAAGCTTTAGGGCACCGCA TAGGCCACCGC-3′). Following *Xba*I/*Hin*dIII digestion, the downstream fragment was cloned into the suicide vector pK18mobsacB. The resulting plasmid was opened with *Eco*RI and *Xba*I and the *puhA-sfYFP2* product cut with the same

enzymes was inserted. This plasmid contained a total insert of 1,428 bp. The fragments were inserted in this order as the *Xba*I site in the OLE-PCR product is blocked by overlapping *dam* methylation once inserted into a plasmid. Conjugative transfer and screening of possible mutants was performed as described for the *crtB* deletion.

**Growth of *Rhodobacter sphaeroides*.** *Rba. sphaeroides* strains were grown in M22 + medium³⁶ supplemented with 10,000X vitamins (0.08 M nicotinic acid, 0.01 M thiamine, 7.3 mM 4aminobenzoic acid, 0.4 mM d-biotin), to a final concentration of 1X. Liquid cultures were supplemented with 0.1% casamino acids. Single colonies were used to inoculate 10 ml starter cultures of M22 medium and grown semi-aerobically in the dark with shaking at 180 r.p.m. at 34 °C for 48 h. For anaerobic photosynthetic cultures, 20 ml of M22 + medium was used to dilute 10 ml of semi-aerobic culture in a 30 ml universal. The culture was incubated for 48 h. This culture was used to inoculate a 1 l Roux bottle filled with M22 + medium and grown for 24 h. The culture was used to inoculate 8 l of M22 + medium and grown to an optical density at 680 nm (OD₆₈₀) of 3. All cultures were gently stirred using magnetic stir bars. Light was provided by 20 W MEGAMAN CFL bulbs at 100 μmol photons s⁻¹ m⁻², measured using a LI-250A Light Meter equipped with a LI-190 Quantum Sensor (LI-COR Biosciences). Where required, cells were pelleted by centrifugation and resuspended in 20 mM HEPES pH 8.

For photosynthetic growth curves, 48 h, 10 ml semi-aerobic starter cultures were diluted to 30 ml and grown photosynthetically for 72 h. Photosynthetically grown cells were used to inoculate 17 ml tubes to a starting OD₆₈₀ of ~0.15. Three tubes were grown per strain and average values and standard deviations are presented (Fig. 3). Light was provided using an array of green LEDs (520 nm emission maximum, 35 nm FWHM, 170 μmol photons s⁻¹ m⁻² (Würth Elektronik, Germany).

**Purification of RCYFP–LH1 complexes.** Monomeric core complexes were purified using a method adapted from ref. 36. Lysozyme was added to the cell sample to a final concentration of 0.5 g l⁻¹ and incubated at 25 °C for 30 min in the dark. A small spatula of deoxyribonuclease 1 from bovine pancreas (Sigma) was added to the cells. The cells were French pressed three times at 18,000 psi. The cells were kept on ice throughout. Unbroken cells were removed by centrifugation at 33,000 g at 4 °C for 25 min. The supernatant was applied to the top of a 15/40% sucrose gradient followed by centrifugation for 16 h at 65,000 g. The ICM fraction was harvested from above the 40 % sucrose layer and pelleted by centrifugation at 125,171 g for 2 h and resuspended in 20 mM HEPES pH 8. The membranes were solubilised with 3% β-dodecyl maltoside (β-DDM) for 1 h at 4 °C with continuous stirring in the dark. The insoluble material was removed by centrifugation at 160,000 g for 1 h. The supernatant containing solubilised membranes was layered on top of discontinuous sucrose gradients containing 20, 21.25, 22.5, 23.75, 25 and 50 % sucrose in 20 mM HEPES, 5 mM EDTA and 0.03 % β-DDM. Gradients were centrifuged at 90,000 g for 40 h. The band containing monomeric RC/YFP–LH1 core complexes was collected.

Core complexes were purified by DEAE ion exchange and eluted with a gradient of 250–400 mM NaCl in 20 mM HEPES, 0.03 % β-DDM, pH 7.8. The core complex eluted at 250 mM NaCl and was subject to further purification using a Superdex 200 gel filtration column equilibrated with 10 mM NaCl 20 mM HEPES, 0.03 % β-DDM, pH 7.8. Fractions having an $A_{873}/A_{280}$ absorbance ratio >1.9 were pooled.

**Purification of RCYFP complexes.** ICMs were solubilised with 1% lauryl dimethyl amine oxide (LDAO) and loaded on to a DEAE Sepharose column equilibrated with 20 mM Tris–HCl, 0.1 % LDAO, pH 7.8. The column was washed with two steps of 50 and 100 mM NaCl and RCs were eluted with 200 mM NaCl. Eluted RCs were pooled and concentrated followed by dilution with 20 mM Tris–HCl, 0.1 % LDAO, pH 7.8. The protein was loaded on to a Q Sepharose (HiTrap QFF) column and eluted with a gradient from 0 to 400 mM NaCl. RC/YFP eluted at 300 mM NaCl and was loaded onto a Superdex 200 (GE Healthcare) gel filtration column equilibrated with 20 mM Tris–HCl, 0.1 % LDAO, 10 mM NaCl. Fractions with an $A_{803}/A_{280}$ nm absorbance ratio of 0.73 or above were pooled.

**Purification of YFP.** The gene encoding SYFP2 was cloned into the vector pBBRBB-$Ppuf_{843-1200}$ (ref. 28) for expression in $\Delta crtB$ *Rba. sphaeroides*. Cells were grown semi-aerobically to an $OD_{680}$ of 3 and disrupted by French pressure treatment. To remove membranes from the soluble fraction, cells were centrifuged at 125,100 $g$ for 2 h and the supernatant was collected. The sample was loaded on to a DEAE Sepharose column equilibrated with 20 mM HEPES, pH 7.8, and eluted with a gradient of 0–400 mM NaCl.. YFP containing fractions were pooled and concentrated using a Centriprep (Amicon) spin concentrator followed by dilution with 20 mM HEPES, pH 7.8. The protein was loaded on to a Q Sepharose (HiTrap QFF) column and eluted with a gradient of 0–400 mM NaCl. YFP eluted at 225 mM NaCl and was loaded onto a Superdex 200 (GE Healthcare) gel filtration column equilibrated with 1 M NaCl, 20 mM HEPES, pH 7.8.

**Immunoblotting.** Protein samples were separated by SDS–PAGE and transferred to a nitrocellulose membrane (Hybond ECL, Amersham) in Transfer buffer (190 mM glycine, 24 mM Tris, 20% (v/v) methanol) at a constant current of 350 mA for 1 h at 4 °C. Following transfer, the membrane was blocked for 1 h in Tris-buffered saline (TBS) (18 mM Tris–HCl pH 7.6 with 68 mM NaCl) with 0.2 % (w/v) Tween-20 and 5 % (w/v) skimmed milk powder, washed in TBS and then incubated with an appropriate dilution of primary antibody in TBS with 0.05 % (w/v) Tween-20 for 4–16 h at room temperature. Anti-RCH[31] and anti-GFP (G1544, Sigma Aldrich) were used at dilutions of 1:5,000 and 1:4,000, respectively. After incubation with primary antibody the membrane was washed with TBS and then incubated with a 1:10,000 dilution of secondary anti-rabbit antibody conjugated with peroxidase (Sigma Aldrich) in TBS. The membrane was washed with TBS before immunodetection using the Amersham ECL Western blotting analysis system (GE Healthcare Life Sciences) according to the manufacturer's instructions. Uncropped immunoblots are displayed in Supplementary Fig. 10.

**Electron microscopy.** Electron microscopy and single particle reconstruction of RC/YFP–LH1 complexes was performed as described previously[37]. The detailed method was as follows. Purified RC/YFP–LH1 complexes were diluted to 0.1 mg ml$^{-1}$. Five µl protein solution was applied to a carbon coated 400 mesh copper EM grid, which was glow discharged 30 s before use. Excess protein solution was blotted by touching the edge of the grid on the surface of Whatman filter paper. After washing twice with distilled water, grids were stained with 0.75% (w/v) uranyl formate for 30 s. The grids were imaged using a Philips CM100 electron microscope operating under 100 kV accelerating voltage at room temperature. All micrographs were recorded with a 1 K × 1 K Gatan Multiscan 974 CCD camera at a normal magnification of × 52,000, corresponding to 3.93 Å per pixel at specimen level with variation of under-focus value from 0.5 to 2.5 µm.

The micrographs were evaluated initially by Digitalmicrograph (Gatan). All images showing drifting, poor staining and unsuitable under-focus were discarded. Selected images were submitted to EMAN2 (ref. 38) for CTF correction and particle picking. A total of 11,777 particles, each 72 × 72 pixels, were treated subsequently using the IMAGIC-5 software package[39]. All particles were initially bandpass filtered to suppress low spatial frequencies. A soft-edged circle mask was used to remove unwanted background of the particle. The masked data were then centered and normalized for reference-free 2D classification, from which a set of characteristic classes was selected for multi-reference alignment (MRA) and multi-variant statistical analysis (MSA) classification. This procedure was iterated until a stable 2D classification was obtained. An initial model was established by assigning Euler angles to the three most different 2D classes selected from the above calculation using angular reconstitution in IMAGIC-5; this model was refined gradually by adding more 2D classes. An anchor set produced from back projection of the initial model was used to calculate the Euler angle of whole 2D class dataset, from which a new 3D model was produced. This procedure was iterated by increasing the calculation precision and decreasing the search step of the Euler angle until a stable 3D model obtained. A new reference produced from this stable 3D model was then used for a new MSA iteration. Finally, 380 2D classes were used for generation of a 3D RC/YFP–LH1 model. The final 2D class dataset was split randomly into two halves, and two 3D models were reconstituted independently. By calculation of Fourier shell correlation (FSC) between these two 3D models, a resolution of 25.9 Å for the final model (Supplementary Fig. 11) was estimated based on 0.5 FSC criteria[40].

**Room temperature and low temperature absorbance spectra.** Room temperature and 77 K absorption spectra were recorded using a Cary ultraviolet/vis spectrophotometer in the spectral range between 260 and 950 nm. For cryogenic measurements, samples suspended in a cryo-stable buffer consisting of 20 mM Tris–HCl, 80% glycerol (v/v) were cooled to 77 K in an Optistat DN-V optical cryostat manufactured by Oxford Instruments.

**Low temperature spectroscopy.** The absorption and excitation fluorescence spectra in Fig. 1 were recorded in a cryo-stable buffer consisting of 20 mM HEPES, 80% glycerol. Samples were cooled to 77 K as described above. Measurements were recorded on a SPEX FluoroLog spectrofluorimeter (SPEX Industries). Excitation was provided from a tungsten light source. Fluorescence excitation spectra were recorded from 400 to 910 nm with an emission of 915 nm, using an average of 10 individual scans. Excitation and emission slit widths of 5 nm were used. Fluorescence excitation spectra were normalized to 1 at 591 nm, data were subject to 2nd order smoothing with 5 neighbours.

**Fluorescence imaging of YFP in *Rhodobacter sphaeroides*.** Cells were washed three times in QH$_2$O then suspended in QH$_2$O. Overall, 15 µl of cell suspension was dropped on 30 µl of 1.5% agar film on a glass slide and sealed with DPX mountant (Sigma − Aldrich) between the glass slide and a coverslip. Fluorescence images were taken with an inverted fluorescence microscope (AxioObserverA1m, Zeiss) equipped with a Hal 100 halogen lamp, a high intensity HBO 100 mercury lamp and an ORCA-ER camera (Hamamatsu). Excitation light was first filtered by a 470/40 nm bandpass filter, then reflected by a 495 nm dichroic beam splitter to the sample through an objective (Plan-Apochromat × 63/1.40 oil objective, Zeiss). Fluorescence emission was filtered by a 520/40 nm bandpass filter before detection by the camera. Each fluorescence image was taken with a 0.1 s exposure time and 50 electron multiplication gain.

**Spectral and lifetime imaging of *Rba. sphaeroides*.** The fluorescence emission properties of YFP in whole cells were measured on a home-built time-resolved fluorescence microscope. The inverted microscope is equipped with a spectrometer (Acton SP2558, Princeton Instruments), an electron-multiplying charge-coupled device (EMCCD) camera (ProEM 512, Princeton Instruments) and a Hybrid Detector (HPM-100-50, Becker & Hickl). The excitation light source was from a pulse supercontinuum white light laser (SC 480-10, Fianium) with a repetition rate of 40 MHz. The laser beam was focused on the sample surface, illuminating a diffraction limited spot using a × 100 objective (PlaneFluorite, NA = 1.4, oil immersion, Olympus). The resulting fluorescence emission was detected through the spectrometer onto the EMCCD camera and the resulting fluorescence lifetime was detected through the spectrometer onto the Hybrid Detector.

During fluorescence spectral and lifetime measurements, the excitation light was filtered by a 472/30 nm bandpass filter, then reflected by a 495 nm dichroic beamsplitter to the sample. The resulting fluorescence emission was filtered by a 496 nm long-pass filter before being detected by the cameras. The fluorescence emission was captured with a slit width of 1.5 mm and a grating of 150 line per mm working at a central wavelength of 550 nm in the spectrometer. Multiple measurements were performed on eight different cells on each sample. Each fluorescence spectrum was detected by EMCCD at an average of three frames with a 1 s exposure time and an electron multiplication gain of 80. Analysis was done with OriginPro.

For fluorescence lifetime measurements, the modulation of the laser was synchronized with a time-correlated single-photon counting (TCSPC) module (SPC-150, Becker & Hickl). Fluorescence lifetimes were recorded by parking the focused laser spot over one single cell, selecting a central wavelength of 550 nm by use of the monochromator and detected by the Hybrid Detector. SPCM software (Becker & Hickl) was used for the data acquisition. The families of decay curves were analysed with OriginPro and TRI2 software packages by fitting with a multiexponential decay function:

$$I(t) = \sum_{i=1}^{n} A_i \exp\left(\frac{-t}{\tau_i}\right) + B \qquad (2)$$

where $\tau_i$ is the fluorescence lifetime, $A_i$ is the fractional amplitude contribution of the $i$th decay component, and $B$ is the background. The quality of fit was judged on the basis of the reduced $\chi^2$ statistic:

$$\chi_{red}^2 = \frac{\sum_{k=1}^{n} \frac{|I(t_k) - I_c(t_k)|^2}{I(t_k)}}{n - p} = \frac{\chi^2}{n - p} \qquad (3)$$

where $I(t_k)$ is the data at time point $k$, $I_c(t_k)$ is the fit at time point $k$, $n$ is the number of data points and $p$ is the number of variable fit parameters ($n$–$p$ = degrees of freedom). The instrument response (IRF) of the system, measured using a mirror, was $\sim 0.18$ ns, and the convolution of the decay curves with the IRF was taken into account when the fitting was performed.

**Optical measurements on purified RC and RCYFP complexes.** Photophysical measurements on RC, YFP or RC/YFP proteins in 0.1% LDAO, 10 mM Tris,

200 mM NaCl, pH 7.8 buffer were carried out at room temperature. Static absorption (Shimadzu UV-1800) and fluorescence measurements (Horiba Nanolog) employed $\sim 1\,\mu M$ samples, which give $A \leq 0.1$ at $\lambda_{exc}$ for fluorescence studies. The latter used 2 nm excitation and emission bandwidths, and spectra were corrected for IRF. Fluorescence quantum yields are the average values obtained using methods: (1) an absolute method using an integrating sphere (Horiba, Quanti-Phi), and (2) a relative method using as a standard fluorescein in 0.1 M NaOH, for which $\Phi_f = 0.91$ was obtained from the average of two literature values[41,42]. Singlet excited-state lifetimes were determined using (1) transient absorption (TA) spectroscopy, (2) time-correlated-single-photon-counting (TCSPC) detection of fluorescence decay (IRF function of $\sim 0.2$ ns), and (3) stroboscopic measurement of fluorescence decay (IRF function of $\sim 1$ ns) as described previously[43]. TA studies utilized an amplified Ti:Sapphire laser system (Spectra Physics) and Helios and Eos detection systems (Ultrafast Systems). Samples were excited with $\sim 0.5\,\mu J$, $\sim 100$ fs 515-nm excitation pulses (at 1 KHz) focused to 1 mm diameter[44]. For RC-only or RC/YFP, a spinning cell (2 mm path and 3 mm wide annulus) containing 3 ml of $\sim 10$–$15\,\mu M$ sample was used to prevent re-excitation of the complex while in the long-lived (1–2 s) $P^+Q_B^-$ state of the RC. A standard 2 mm path cuvette with stirring was used for YFP-only (600 µl at $\sim 15\,\mu M$).

Förster calculations were performed using the measured yield of excitation energy transfer ($\Phi_{EET}$) from excited YFP to RC to obtain the donor to acceptor distance ($R$) between these units. The calculations employed PhotochemCAD[32], which uses the '$R_0$ method' in which $\Phi_{EET} = R_0^6/(R_0^6 + R^6)$, where $R_0^6 = (8.8 \times 10^{23})$ $\kappa^2\,\Phi_f\,J\,n^{-4}$ and $R_0$ is the distance at which $\Phi_{EET} = 0.5$. Here, the orientation factor $\kappa^2 = 0.667$ assuming a random orientation of transition dipoles, $\Phi_f = 0.70$ (vide infra) is the donor (YFP) fluorescence yield, $J$ is the spectral-overlap factor (a molar absorptivity of $288{,}000\,mM^{-1}\,cm^{-1}$ at 800 nm was used for the RC[45]), and $n$ is the refractive index, for which values of 1.335, 1.45 and 1.55 were used to accommodate protein/cofactor environments ranging from buffer to pure hydrocarbon (Supplementary Figs 5 and 8).

**Data availability.** The authors declare that the data supporting the findings of this study are available within the paper and its Supplementary Information files or are available from the corresponding author upon request.

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

## Acknowledgements

This work was supported as part of the Photosynthetic Antenna Research Center (PARC), an Energy Frontier Research Center funded by the U.S. Department of Energy, Office of Science, Office of Basic Energy Sciences under Award Number DE-SC 0001035. PARC supported photophysical studies of RC/YFP, YFP and RC complexes (P.D., J.M.Y., D.M.N., C.K., K.J.G.), and fluorescence lifetime studies of whole cells (X.H., C.V.) with support for X.H., P.D., C.V., Y.M.M., D.M.M., and partial support for C.N.H. C.N.H. also acknowledges financial support from Advanced Award 338895 from the European Research Council. C.N.H. and A.H. acknowledge financial support from the Bio-technology and Biological Sciences Research Council (BBSRC UK), award number BB/G021546/1. K.J.G. was supported by a doctoral studentship from the Engineering and Physical Sciences Research Council (EPSRC UK), and gratefully acknowledges a Scientific Exchange award from PARC; G.J.L. thanks the EPSRC (Grant EP/I012060/1) for financial support. K.M.F. was supported by the National Science Foundation Graduate Research Fellowship under grant DGE-1143954. The authors are grateful to Dr Amanda Brindley and Dr David Mothersole for useful discussions.

## Author contributions

C.N.H. conceived the project and designed the experiments. K.J.G., C.V., K.M.F, D.H. and C.K. assisted in the experimental design. K.J.G., K.M.F., X.H., P.Q., P.D., E.C.M., A.H., J.M.Y., D.M.N. and C.K. performed the experiments. P.Q. produced the RC/YFP model. K.M.F. and X.H. provided analysis of spectroscopic data. C.N.H. and G.J.L. supervised the project. C.N.H., A.H., K.J.G., D.H., C.K., and K.M.F. wrote the manuscript. All authors were involved in discussion of the results and revision of the manuscript.

## Additional information

**Competing financial interests:** The authors declare no competing financial interests.

