## [Peer Review File · Nature Communications]

Reviewers' comments:

Reviewer #1 (Remarks to the Author):

Grayson et al report the production and characterisation of a mutant of the purple photosynthetic bacterium *Rhodobacter sphaeroides* with a YFP tag attached to the reaction centre. They demonstrate very convincingly that light absorbed by the YFP is transferred to the photosynthetic pigments with a quantum efficiency of around 40%, and furthermore that YFP-absorbed light significantly enhances phototrophic growth under appropriate illumination conditions. This is a very striking finding - I would have been very sceptical about the idea that augmenting the reaction centre with a single extra chromophore could measurably enhance phototrophic growth, but clearly it does. The study acts as a very nice proof of principle for an *in vivo* approach to tailoring photosynthesis for optimal efficiency in different light environments. The work is very thorough and clearly presented. I can find just one point that needs clearing up:

Growth assays were performed with white-light illumination at 100 microeinsteins (p.4) or possibly 150 microeinsteins (p.15), or with green LED illumination 170 microeinsteins (p.15). The authors should comment on the white and green light intensities required to saturate photosynthesis and phototrophic growth in the background strain. Obviously, augmenting light-harvesting can only be effective for promoting growth at sub-saturating light intensities, with the magnitude of the effect increasing as light becomes more limiting for growth. Therefore to put the growth effects seen in Fig. 2 into context, it's important to know the extent of light limitation under these experimental conditions.

Reviewer #2 (Remarks to the Author):

This paper is detailed and comprehensive. The work reported includes genetic modifications of purple photosynthetic bacteria to incorporate a complementary light absorber, and the action of that change is elucidated. The work is not trivial and there are elements of the work that reveal the deep thought behind the research. For example, since the YFP absorption is obscured by carotenoids, a carotenoid-lacking mutant is used. However, that leads to the complication that the light harvesting antenna complexes are not assembled. A series of appropriate and systematic controls are used therefore to assess how the modified bacteria harvest light and whether the new chromophore helps. Careful photophysical studies determine that the energy transfer is about 50% from the YFP to the reaction center. Cryo-EM was used to determine the structure of the model. Overall there is an astounding amount of work reported.

In conclusion, the results do not provide a revolutionary example of biomodified light harvesting, but they do make important contributions by showing that it is possible to augment photosynthetic light harvesting, and that future work will need to employ advanced design of the artificial light harvesters--intrinsic chromophore proteins like YFP are convenient, but not well suited for efficient light harvesting. Overall this is a significant

contribution.

Reviewer #3 (Remarks to the Author):

In the manuscript of "Augmenting light coverage for photosynthesis through YFP-enhanced charge separation at the Rhodobacter sphaeroides reaction centre", Grayson et al. used the negative-staining electron microscope and single-particle 3D reconstruction methods studied the 3D structure of the RC/YFP-LH1 complex. The EM images are of good resolution and the 3D reconstruction is of reasonable quality. My minor comments are below

1. Line, 164, "Purified RC/YFP-LH1 complexes were adsorbed onto copper grids". The sample cannot be adsorbed on the copper grid, but adsorbed on the pre-coated carbon film on the copper grid.
2. Line 166, Fig. 3b shows examples of averaged 2D classes. Author should clear to state the averages are referenced class averages or reference-free averages.
3. Author should include the detail experimental information in the method instead of just a simple citation. The cited paper do not have sufficient information either, such as how the "selected 2D averaged classes were used to establish the initial 3D model"?
4. The single particle 3D reconstruction is significant depended on the given initial model; author should state how the initial model was generated and how to avoid the potential initial model bias during the 3D reconstruction.
5. Authors should show the FSC analysis result and resolution determination criteria, based on frequency at 0.5 or 0.143?
6. Authors should discuss the potential artifact from negative-staining, why cryo-EM is not the solution for this study.

Reviewer #1:

Growth assays were performed with white-light illumination at 100 microeinsteins (p.4) or possibly 150 microeinsteins (p.15), or with green LED illumination 170 microeinsteins (p.15).

We apologise for the error; the white-light illumination was 100 microeinsteins, now corrected on line 472. The figure for green light illumination is correct.

The authors should comment on the white and green light intensities required to saturate photosynthesis and phototrophic growth in the background strain. Obviously, augmenting light-harvesting can only be effective for promoting growth at sub-saturating light intensities, with the magnitude of the effect increasing as light becomes more limiting for growth. Therefore to put the growth effects seen in Fig. 2 into context, it's important to know the extent of light limitation under these experimental conditions.

This is a good point, which we have addressed partly by including some new text and also by inclusion of some extra data.

Fig. 2a now shows a positive control (green points) that demonstrates how quickly cells can grow under these white light conditions when carotenoid pigments are present. Thus, a comparison of the carotenoid-containing and carotenoid less strains provides a measure of how 'underemployed' the RCs are when carotenoids are missing. The growth rate for the $\Delta crtB$ strain is approximately half the control rate, so this strain grows in sub-saturating light. New text on lines 112-147 explains this:

'The LH2-minus positive control (carotenoids+RC-LH1-PufX), chosen because the other two strains also lack LH2 due to the $\Delta crtB$ mutation, shows strong photosynthetic growth (Fig. 2a, green). Allowing for the ~30 hour lag delaying the onset of photosynthetic growth the subsequent difference between the green and blue growth curves shows that under white light illumination the $\Delta crtB$ culture is light-limited by approximately 50 %. This effect arises because the absence of carotenoids impairs the light-absorbing capacity of the LH1 antenna, thereby limiting turnover at the RCs.'

We also include new text on lines 151-157 that explains the issue of light limitation when green LEDs are used:

'The difference between the positive control (carotenoids+RC-LH1-PufX; green line) and $\Delta crtB$ strains (no carotenoids, red/blue lines) in Fig 2c shows that both $\Delta crtB$ and $\Delta crtB$ RC/YFP-LH1 are growing at sub-saturating light intensities. Illumination of carotenoid-less cultures by green LEDs barely allows native BChl pigments to contribute towards light harvesting. Comparison of the green and blue growth curves in Fig. 2c suggests that RC turnover in the $\Delta crtB$ strain runs at only ~10 % of its potential capacity so the addition of a 520 nm-absorbing chromophore to the $\Delta crtB$ strain might be expected to be beneficial, as long as absorbed energy can transfer to photosystem components.'

Reviewer #2:

This paper is detailed and comprehensive. The work reported includes genetic modifications of purple photosynthetic bacteria to incorporate a complementary light absorber, and the action of that change is elucidated. The work is not trivial and there are elements of the work that reveal the deep thought behind the research. For example, since the YFP absorption is obscured by carotenoids, a carotenoid-lacking mutant is used. However, that leads to the complication that the light harvesting antenna complexes are not assembled. A series of appropriate and systematic controls are used therefore to assess how the modified bacteria harvesting light and whether the new chromophore helps. Careful photophysical studies determine that the energy transfer is about 50% from the YFP to the reaction center. Cryo-EM was used to determine the structure of the model. Overall there is an astounding amount of work reported.

In conclusion, the results do not provide a revolutionary example of biomodified light harvesting, but they do make important contributions by showing that it is possible to augment photosynthetic light harvesting, and that future work will need to employ advanced design of the artificial light harvesters--intrinsic chromophore proteins like YFG are convenient, but not well suited for efficient light harvesting. Overall this is a significant contribution.

We thank the Reviewer for these comments.

Reviewer #3:

In the manuscript of "Augmenting light coverage for photosynthesis through YFP-enhanced charge separation at the Rhodobacter sphaeroides reaction centre", Grayson et al. used the negative-staining electron microscope and single-particle 3D reconstruction methods studied the 3D structure of the RC/YFP-LH1 complex. The EM images are of good resolution and the 3D reconstruction is of reasonable quality. My minor comments are below.

1. Line, 164, "Purified RC/YFP-LH1 complexes were adsorbed onto copper grids". The sample cannot be adsorbed on the copper grid, but adsorbed on the pre-coated carbon film on the copper grid.

Now corrected on line 199.

2. Line 166, Fig. 3b shows examples of averaged 2D classes. Author should clear to state the averages are referenced class averages or reference-free averages.

Now corrected on line 202.

3. Author should include the detail experimental information in the method instead of just a simple citation. The cited paper do not have sufficient information either, such as how the "selected 2D averaged classes were used to establish the initial 3D model"?

Full details are now provided in the Methods section on lines 533-558, and three more references (38-40) were added to the list.

4. The single particle 3D reconstruction is significant depended on the given initial model; author should state how the initial model was generated and how to avoid the potential initial model bias during the 3D reconstruction.

A statement is provided on lines 548-550. If there were no pre-existing structural data for the WT complex without YFP, then potential initial model bias could arise. However, we have worked on the structural of the RC-LH1 complex, using negative stain EM and 3D crystallography, for 16 years and can readily recognise the extra density for YFP, which is clearly visible even in some raw images as shown in the inset to Fig. 3a.

5. Authors should show the FSC analysis result and resolution determination criteria, based on frequency at 0.5 or 0.143?

The calculation of Fourier shell correlation (FSC), showing a resolution of 25.9 Å for the final model, (see Supplementary Fig. 9) was estimated based on a frequency of 0.5.

6. Authors should discuss the potential artifact from negative-staining, why cryo-EM is not the solution for this study.

We now include the following text on lines 430-437:

‘The use of negative stain limited the resolution of the structural model to ~26 Å, but the presence of YFP was evident from the outset, even in raw images, and the final structural model clearly shows a density that can be confidently assigned to YFP. The staining procedure could affect the perceived orientation of YFP with respect to the rest of the complex; we used over 11,000 particles for calculating the structural model so Fig. 3 shows an averaged orientation of YFP. The use of cryo-EM would produce a better quality structural model, which would also allow a more precise estimate of the distance between the YFP chromophore and the RC special pair. Nevertheless, the low resolution data are sufficient for a calculation of an ~85 Å distance.....’

We realise that cryo-EM would produce a better structural model, but we were limited by the time and facilities available. Given that we had also estimated the YFP-RC energy transfer distance using a variety of spectroscopic methods, this structural confirmation of proximity between RC and YFP was not the main focus of the paper.

REVIEWERS' COMMENTS:

Reviewer #1 (Remarks to the Author):

The authors have done a reasonable job of answering my major point about the extent to which light is limiting for growth under their illumination conditions. They haven't answered it in exactly the way I was expecting, which would have been by providing light saturation curves for growth. However, I appreciate that these may be difficult to produce if the light intensity can't be increased enough to get to the top end of the curve. Instead they have used the growth of a carotenoid-containing strain as a control for what happens when you increase antenna size. This isn't completely ideal, and if I was being really picky I could suggest that slower growth in the carotenoid-deficient strains could be for reasons other than reduced antenna size (perhaps destabilisation of the photosynthetic apparatus or susceptibility to photodamage). However, I am satisfied that increased antenna size in the YFP-tagged strain is the only reasonable explanation for the fact that it grows faster than the background strain, and the authors' new data and comments do at least raise the issue of light-saturation for consideration by the reader.

Reviewer #3 (Remarks to the Author):

Authors have addressed all my questions in the revised manuscript. This reviewer has no more questions and recommend it for publication.